

# A new small device made of glass for separating microplastics from marine and freshwater sediments

Ryota Nakajima, Masashi Tsuchiya, Dhugal J. Lindsay, Tomo Kitahashi, Katsunori Fujikura and Tomohiko Fukushima

Japan Agency for Marine-Earth Science and Technology (JAMSTEC), Yokosuka, Kanagawa, Japan

## ABSTRACT

Separating microplastics from marine and freshwater sediments is challenging, but necessary to determine their distribution, mass, and ecological impacts in benthic environments. Density separation is commonly used to extract microplastics from sediments by using heavy salt solutions, such as zinc chloride and sodium iodide. However, current devices/apparatus used for density separation, including glass beakers, funnels, upside-down funnel-shaped separators with a shut-off valve, etc., possess various shortcomings in terms of recovery rate, time consumption, and/or usability. In evaluating existing microplastic extraction methods using density separation, we identified the need for a device that allows rapid, simple, and efficient extraction of microplastics from a range of sediment types. We have developed a small glass separator, without a valve, taking a hint from an Utermöhl chamber. This new device is easy to clean and portable, yet enables rapid separation of microplastics from sediments. With this simple device, we recovered 94–98% of <1,000 μm microplastics (polyethylene, polypropylene, polyvinyl chloride, polyethylene terephthalate, and polystyrene). Overall, the device is efficient for various sizes, polymer types, and sediment types. Also, microplastics collected with this glass-made device remain chemically uncontaminated, and can, therefore, be used for further analysis of adsorbing contaminants and additives on/to microplastics.

# INTRODUCTION

Microplastics, small pieces of plastic ranging from five mm in size down to microscopic, are ubiquitously distributed particulate contaminants that have been to-date detected in various environmental samples, including freshwater lakes and rivers (*Eriksen et al., 2013*; *Free et al., 2014*; *Gasperi et al., 2014*; *Biginagwa et al., 2016*), seas and oceans, including the deep-sea and the polar regions (*Lusher et al., 2014*; *Shim & Thompson, 2015*; *Bergmann et al., 2017*; *Matsuguma et al., 2017*; *Wang et al., 2019*). Once they end up in aquatic environments, microplastics eventually accumulate in/on the surface water, shores/beaches, and in the sediments of benthic environments (*Hidalgo-Ruz & Thiel, 2013*; *Van Cauwenberghe et al., 2013*; *Law & Thompson, 2014*; *Van Sebille et al., 2015*). Marine and freshwater sediments are considered to possibly be the biggest sink of plastics in

Corresponding author
Ryota Nakajima,
nakajimar@jamstec.go.jp

aquatic environments since significant numbers of microplastic fragments, films, spheres, and fibers have been found in multiple sediment samples from various locations (*Woodall et al., 2014*; *Bergmann et al., 2017*; *Matsuguma et al., 2017*). However, we have not yet reached a consensus about the amounts of microplastics accumulating in oceanic and freshwater sediments, how these microplastics were transported into the sediments, and what their impact is on benthic animals (*Farrell & Nelson, 2013*; *Ugolini et al., 2013*; *Law & Thompson, 2014*; *Green et al., 2016*). Increased amounts of microplastics in/on the sediment enhance the risk of microplastic ingestion by benthic animals (*Van Cauwenberghe et al., 2015a*). Developing an efficient method for detecting microplastics from sediments is an important issue for further understanding the distribution, mass, and ecological impacts of microplastics in both marine and freshwater environments.

One of the major concerns in microplastic studies of sediment samples is methodological—what is the best way to extract microplastics from a large amount of sediments? Several methods have been proposed to extract microplastics from sediment samples, including visual sorting, filtration, sieving, density separation, elutriation, flotation, chemical digestion, electrostatic behavior, and magnetic extraction (*Hidalgo-Ruz et al., 2012*; *GESAMP, 2015*; *Masura et al., 2015*; *Duis & Coors, 2016*; *Felsing et al., 2018*; *Grbic et al., 2019*). Density separation is a commonly used method to extract microplastics from sediments or sand using the principle of the difference in specific density of sediments (e.g., 2.6 g cm$^{-3}$) and plastics (0.1–1.7 g cm$^{-3}$) (*Chubarenko et al., 2016*). In density separation, dense solutions that have been proposed include sodium polytungstate (density: ~3.2 g cm$^{-3}$), zinc chloride (ZnCl$_2$, density: ~1.8 g cm$^{-3}$), zinc bromide (ZnBr$_2$, density: 1.7 g cm$^{-3}$), and sodium iodide (NaI, density: ~1.8 g cm$^{-3}$) (*Corcoran, Biesinger & Grifi, 2009*; *Imhof et al., 2012*; *Liebezeit & Dubaish, 2012*; *Dekiff et al., 2014*; *Van Cauwenberghe et al., 2015b*; *Quinn, Murphy & Ewins, 2017*; *Prata et al., 2018*). Due to the higher densities of these salt solutions compared to most microplastics (0.9–1.5 g cm$^{-3}$), less dense microplastics float and separate out from the more dense, sinking, sediment materials (*Quinn, Murphy & Ewins, 2017*).

During density separation, a glass beaker is often used, but adhesion of microplastics onto the glass wall is a problem when pouring or sucking supernatant containing microplastics, thus requiring repeating the procedure. Avoiding resuspension of decanted sediments is also challenging when pouring supernatants (*Masura et al., 2015*). Use of a glass funnel is thus employed to avoid resuspension of sediments, but only a very small amount of sediment can be processed at any one time, requiring repeating the procedure (*Masura et al., 2015*). There is an upside-down funnel-shaped microplastic separator with a shut off valve (MPSS), but it is a large and heavy metal device and is thus not portable (*Imhof et al., 2012*). It also requires dismantling of the unit, including the shut off valve, for cleaning purposes between samples (*Imhof et al., 2012*). There is also a small and portable device based on the above large metal device (SMI), but it still requires cleaning of the shut off valve between samples (*Coppock et al., 2017*). Also, separators made of plastics (such as SMI) may not be appropriate for further analysis of adsorbing contaminants and additives on/to microplastics. In evaluating existing microplastic extraction devices using density separation, we identified the need for a device that allows
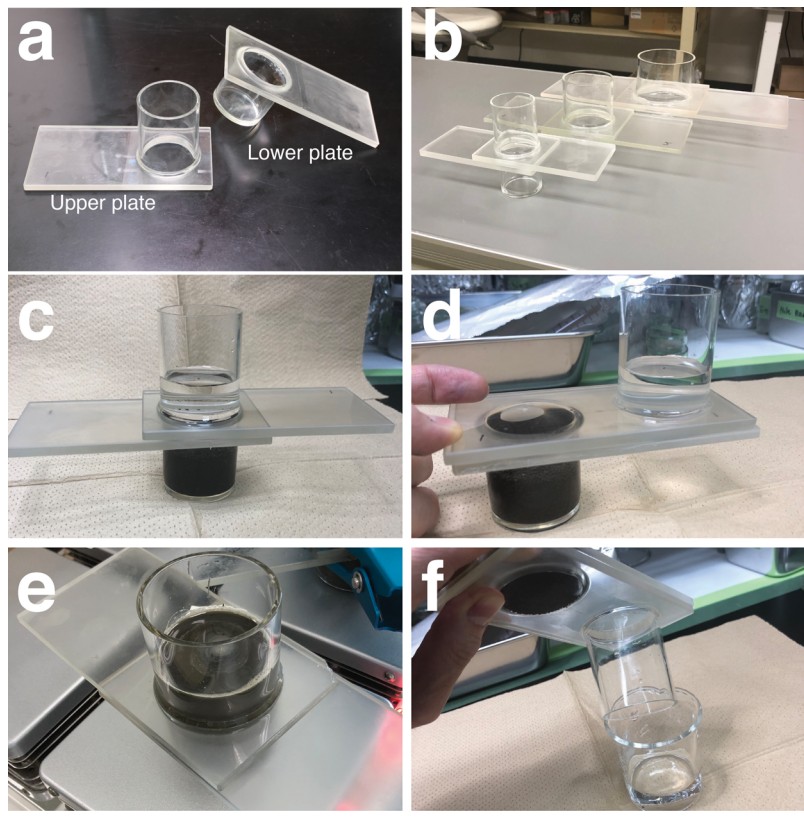

**Figure 1 JAMSTEC microplastic-sediment separator (JAMSS) unit.** (A) The upper plate (left) incorporates an open glass tube, while the lower plate (right) incorporates a cylindrical glass container. (B) Small, middle, and large models of assembled JAMSS, consisting of a cylindrical container of 30, 60, and 100 ml volume, respectively. (C) JAMSS during density flotation with sediment in the lower container. (D) Separation of sediment and supernatant by sliding the two plates against each other. (E) JAMSS can be placed on a magnetic stirrer to ensure the sediments are well mixed during microplastic flotation. (F) Microplastics in the supernatant in the upper tube are poured by rinsing the internal walls of the tube.

rapid, simple, and efficient extraction of microplastics from a range of sediment types. Here, we report a simple and small device made of glass, without a valve, for easier separation of microplastics from sediments.

## MATERIALS AND METHODS

We developed a small separation unit made of glass without a valve, based loosely around a Utermöhl chamber (*Utermöhl, 1931*), that can isolate microplastics from sediments in a single step. The JAMSTEC microplastic-sediment separator (JAMSS) unit is constructed of two components—one upper and one lower modified glass plate (Fig. 1A). JAMSS can handle a variety of different sediment amounts by changing the volume of the tubes/containers attached to the lower and upper plates. Figure 1B shows small, middle, and large models of JAMSS, consisting of a cylindrical container of 30, 60, and 100 ml volume, respectively. The upper plate incorporates an open glass tube, while the lower plate incorporates a cylindrical glass container (Fig. 1A). Each glass plate is frosted (ground) on the opposite side to that from which the respective tube or container is attached.

The two plates are set against each other during use, initially so that the lower container and upper tube become a single, cylindrical container into which the sample can be poured (Fig. 1C). The plates can then be slid against each other to separate the lower and upper halves of the sample, respectively, by introducing a lid and bottom (the glass plate) to each half (Fig. 1D). The two plates stick together through the addition of a liquid to the frosted glass surface. Silicone grease can also be applied to the longitudinal edges of the glass plates to help them stick together. Metal clips can be attached to the plates for a secure attachment when the plates are heavy/large (e.g., large plate in Fig. 1B).

## Microplastic separation from sediments

With the JAMSS unit in its initial position, the internal walls of cylinders were thoroughly rinsed with distilled water prior to use. A sediment sample was placed in the lower cylindrical container using a spatula. NaI solution was prepared by dissolving the salt in distilled water to achieve a final density of $1.6$ g cm$^{-3}$. We used NaI solution in this study because it is less environmentally dangerous compared to other commonly used salt solutions such as $ZnBr_2$ (*Quinn, Murphy & Ewins, 2017*). The solution was filtered through glass fiber filters (Whatman GF/F) prior to use in order to remove any contaminants. Then, the filtered NaI solution was poured into the JAMSS unit so that the solution filled the cylinder to at least halfway up the upper tube. A glass-covered magnetic stirring bar (VWR) was used to gently mix the sediment for 20 min (Fig. 1E), and then the unit was left to settle until the supernatant was clear of sediment. Next, the two plates were carefully slid so that the lower tubular container was completely sealed by the upper plate, ensuring there was no resuspension of sediments into the upper tube. The supernatant in the upper tube was then poured and filtered through a filter (such as a PTFE filter) by rinsing the internal walls of the tube with a squirt bottle filled with Milli-Q water (Millipore, Burlington, MA, USA), in order to transfer all residual microplastics onto the filter (Fig. 1F). Filters were then transferred to a clean glass petri dish with a glass cover for further counting of microplastics. All the above procedures took place in a laminar flow hood, with the JAMSS unit being covered with aluminum foil when settling the sediment to minimize contamination.

## Validation of JAMSS

To evaluate the extraction efficiencies of the JAMSS unit, first we tested the unit using finer sediments with known concentrations of microplastics of different polymer types and particle sizes. Secondly, we tested the unit using different sizes of sediment particles.

For the first experiment, we prepared five polymer types: polyethylene (PE, $0.92$–$0.97$ g cm$^{-3}$), polypropylene (PP, $0.90$–$0.91$ g cm$^{-3}$), polystyrene (PS, $0.04$–$1.09$ g cm$^{-3}$), polyvinyl chloride (PVC, $1.35$–$1.45$ g cm$^{-3}$), and polyethylene terephthalate (PET, $1.34$–$1.39$ g cm$^{-3}$). These polymers were prepared in three size categories: 100–500, 500–1,000, and 1,000–2,000 μm by milling plastic plates with a plastic-grinder (PLC-2M; Osaka Chemical, Osaka, Japan) and then size-fractionated using stainless steel mesh screens of 100, 500, 1,000, and 2,000 μm mesh size. We chose these three size categories as they are major size classes of microplastics found in sediments from

various locations (*Maes et al., 2017*; *Matsuguma et al., 2017*; *Peng et al., 2018*; *Mu et al., 2019*; *Zhang et al., 2019*). We used fine-grained aquarium sand (ADA; Amazonica, Miami, FL, USA) as sediment, and this was ground beforehand using a mortar and pestle after adding Milli-Q water to moisten the sand to ease the grinding process. Prepared microplastics (40 particles per replicate) were mixed with 30 g sand (wet weight) in the unit and filtered NaI solution was then added. These added microplastics were distinguishable from the contaminants in the sand (mostly fibers), ensuring that only the added plastics were counted during the final enumeration. In total, five replicates of microplastic-sediment samples were examined for each plastic type and size. Microplastic-sediment samples were stirred in the JAMSS unit with a glass-covered magnetic stirrer as described above. Following filtering onto a membrane filter, the microplastics were counted under a stereomicroscope.

For the second experiment, we prepared reef carbonate sediments (Platinum reef sand; JUN Company Limited, Tokyo, Japan) in three size categories: fine ($31.9 \pm 93.8$ μm), medium ($533.8 \pm 119.1$ μm), and coarse ($1,165 \pm 212$ μm). All the sediments were soaked in the NaI solution prior to use to remove possible contamination. PVC microplastics with a specific gravity of 1.35 (40 particles, size 500–1,000 μm) were mixed with each sediment sample in the unit (10 g dry weight) and filtered NaI solution was then added. In total, five replicates of PVC microplastic-sediment samples were examined for each sediment type, as described above.

The differences in the extraction efficiencies of microplastics between different polymer types or different particle sizes were determined using one-way ANOVA. The normality of the data and homogeneity of variance were examined and verified before ANOVA analysis using a chi-square test and a Bartlett test, respectively. When the one-way ANOVA was significant, differences among means were analyzed using Tukey–Kramer multiple comparison tests. A difference at $P < 0.05$ was considered significant.

## RESULTS AND DISCUSSION

Microplastics artificially incorporated into fine sand were extracted using the JAMSS unit with NaI at a density of 1.6 g cm$^{-3}$. No significant differences were found between the mean recovery rates of each different polymer type in any of the size categories ($P = 0.57$ for 100–500 μm particles; $P = 0.25$ for 500–1,000 μm particles; $P = 0.06$ for 1,000–2,000 μm particles, one-way ANOVA) (Fig. 2). When comparing the recovery rates of microplastics between the three size categories (i.e., 100–500, 500–1,000, 1,000–2,000 μm), overall mean recovery rates slightly decreased with decreasing size of the microplastic particles ($P < 0.0005$, one-way ANOVA) with significant differences between 100–500 and 500–1,000 μm particles and 100–500 and 1,000–2,000 μm particles ($P < 0.05$ Turkey–Kramer). These differences were probably because of the increased chance to lose smaller particles during processing of the samples (*Nakajima et al., 2019*). The recovery rates ranged from $92.5\% \pm 5.6\%$ to $96.0\% \pm 4.2\%$ for 100–500 μm microplastics (overall mean $94.0\% \pm 1.5\%$), $96.5\% \pm 2.9\%$ to $100\% \pm 0\%$ for 500–1,000 μm microplastics (overall mean $97.8\% \pm 1.3\%$), and $97.8\% \pm 1.6\%$ to $100\% \pm 0\%$ for 1,000-2,000 μm microplastics (overall mean $99.1\% \pm 1.0\%$). The recovery rates for <1,000 μm microplastics (94.0–97.8%)

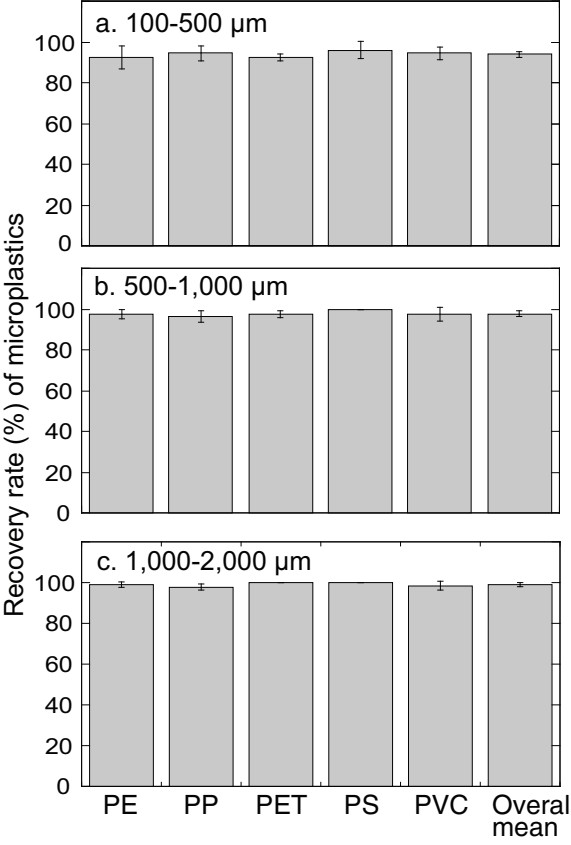

**Figure 2 Recovery rate (%) of microplastics of different polymer types and particle sizes by JAMSS unit.** Each bar indicates the average recovery rates of microplastics of different size particles: (A) 100–500 μm. (B) 500–1,000 μm. (C) 1,000–2,000 μm. PP, polypropylene; PE, polyethylene; PET, polyethylene terephthalate; PS, polystyrene; PVC, polyvinyl chloride. Error bars indicate standard deviation.                                                

were comparable with those of previously developed separators, 95.5% for MPSS (*Imhof et al., 2012*) and 95.8% for SMI (*Coppock et al., 2017*).

Microplastics incorporated into different granularities of reef sediment (fine, 32 μm; medium, 534 μm; coarse, 1,165 μm) revealed the JAMSS unit extracted microplastics efficiently (96.0% ± 3.4% for fine sand, 98.5% ± 1.4% for medium sand, 97.0% ± 1.1% for coarse sand) (Fig. 3). No significant differences ($P = 0.23$, one-way ANOVA) were found between the three grain sizes, showing efficient microplastic separation out from sediments using heavy salt solution (NaI), regardless of grain size.

Among the previous methods for density separation of microplastics, the classical decanting method, for example, the use of a beaker, is simple in design but adhesion of microplastics to the inside of the container is a problem when the media is transferred, thus resulting in a relatively low recovery rate (40%) (*Imhof et al., 2012*). The technique is often repeated three to five times to raise the extraction efficiency, but this extends the processing time for each sample and increases the chance to lose microplastic samples (*Nakajima et al., 2019*). Avoiding resuspension of decanted sediments is also challenging when pouring supernatant (*Masura et al., 2015*). Conversely, JAMSS completely seals

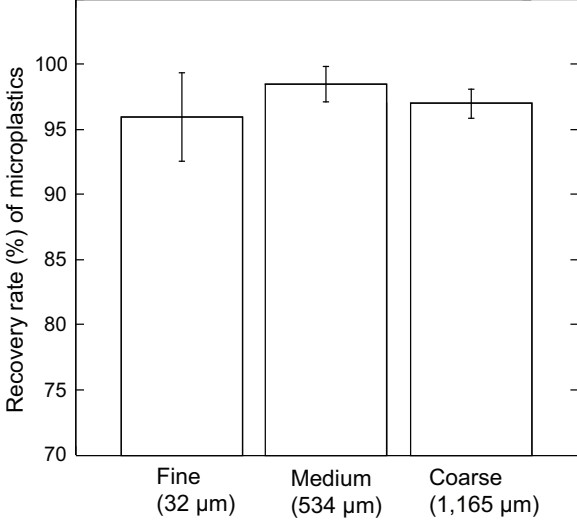

**Figure 3 Recovery rate (%) of microplastics by JAMSS unit using different sizes of sediment particles.** Each bar indicates percentage recovery rate of microplastics in different granularities of reef sediment (fine, medium, coarse). Error bars indicate standard deviation.

the sediment in a closed space (i.e., the lower container) therefore stopping the introduction of resuspended sediments into the supernatant. The microplastic-containing supernatant can then be easily transferred by rinsing the inside walls of the unit, allowing high extraction efficiency of microplastics in a single step.

To combat the weak points of the classical decanting method, use of a glass funnel can be employed to avoid resuspension of sediments, but only a very small amount of sediment can be processed at any one time, requiring repeating the procedure (*Masura et al., 2015*). The device is also easily clogged irrevocably when processing relatively large, coarse sediments. In contrast, JAMSS can handle a variety of different sediment amounts by changing the volume of the tubes/containers attached to the lower and upper plates (Fig. 1B). Currently, the large model of JAMSS consists of a cylindrical container of 100 ml volume, allowing direct placement of the entire sediment content of a core sample slice (e.g., 10 mm thickness, 80 mm diameter).

Previously, an upside-down funnel-shaped MPSS had been proposed as the best method for separating microplastics from sediment samples, but this is a large and heavy metal device and is thus not portable (*Imhof et al., 2012*). It also requires dismantling of the unit, including the shut off valve, for cleaning purposes between samples (*Imhof et al., 2012*). There is also a small, portable device based on the above large metal device (SMI), but it still requires cleaning of the shut off valve between processing samples (*Coppock et al., 2017*). From this point of view, JAMSS has no valve and has a simple structure, allowing rapid cleaning, and reductions in sample processing times.

Though SMI is a portable unit (*Coppock et al., 2017*), it is made of plastic (PVC), thus it may not be appropriate for further analysis of adsorbing persistent organic pollutants and other chemicals such as additives on/to microplastics. One of the advantages of the

JAMSS is that it is made of glass, so microplastics remain uncontaminated for use in further chemical analyses.

Overall, the JAMSS unit is a small, portable device that enables the extraction of microplastics from sediments in a single step. It is simple, without a shut-off valve, and is thus easy to use. The JAMSS has also proven compatible with both finer sediments and coarse sand. Because of the size of this small device, all the above procedures can be done in a laminar flow hood. It should be noted, however, that when the sediment is clay-like or otherwise interferes with the rotation of the magnetic stirrer bar, dilution with water may be necessary. Microplastics can also get caught in the silicone grease used for lubrication between the two plates, so the grease should only be applied to the longitudinal edges of the plates, which do not come into contact with the sample.

## CONCLUSIONS

We have developed a small, glass JAMSS without a shut off valve, taking a hint from an Utermöhl chamber. This new device is easy to clean and portable, yet enables rapid separation of microplastics from sediments in a single step. In using the JAMSS unit, the user has the advantage of being able to rinse the entire headspace multiple times without resuspending the settled sediment, therefore reducing the need for repetitive processing and limiting the possibilities of external contamination and loss of microplastics. The lack of a shut off valve means there is no chance of it becoming clogged. Unlike devices made of plastic, microplastics separated from sediments with this glass-made JAMSS unit remain chemically uncontaminated so they can be used for further analysis of adsorbing contaminants and additives on/to microplastics.

## ACKNOWLEDGEMENTS

We thank Rie Matsui for her help in validating the unit.

### Funding

This study was supported by the Environmental Research and Technology Development Fund (S II-2) of the Environmental Restoration and Conservation Agency of Japan and by the Ocean Resource Use Promotion Technology Development Program of the MEXT, Japan. The funders had no role in study design, data collection and analysis, decision to publish, or preparation of the manuscript.

### Grant Disclosures

The following grant information was disclosed by the authors:
Environmental Research and Technology Development Fund (S II-2) of the Environmental Restoration and Conservation Agency of Japan.
Ocean Resource Use Promotion Technology Development Program of the MEXT, Japan.

### Competing Interests

The authors declare that they have no competing interests.

## Author Contributions

- Ryota Nakajima conceived and designed the experiments, performed the experiments, analyzed the data, contributed reagents/materials/analysis tools, prepared figures and/or tables, authored or reviewed drafts of the paper, approved the final draft.
- Masashi Tsuchiya conceived and designed the experiments, authored or reviewed drafts of the paper, approved the final draft.
- Dhugal J. Lindsay conceived and designed the experiments, authored or reviewed drafts of the paper, approved the final draft.
- Tomo Kitahashi analyzed the data, contributed reagents/materials/analysis tools, authored or reviewed drafts of the paper, approved the final draft.
- Katsunori Fujikura conceived and designed the experiments, contributed reagents/materials/analysis tools, authored or reviewed drafts of the paper, approved the final draft.
- Tomohiko Fukushima analyzed the data, contributed reagents/materials/analysis tools, authored or reviewed drafts of the paper, approved the final draft.

## Data Availability

The raw measurements are available as a Supplemental File.

## Supplemental Information

Supplemental information for this article can be found online at http://dx.doi.org/10.7717/peerj.7915#supplemental-information.

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
