# Peer review of "A new small device made of glass for separating microplastics from marine and freshwater sediments"

_PeerJ, doi:10.7717/peerj.7915_

## Round 0.1 · original submission · Major Revisions

Please carefully consider and address the comments from both reviewers.

Reviewer 1 ·

Basic reporting

The authors Nakajima et al. developed an extraction device to recover microplastic particles from sediment samples. Microplastics are a ubiquitously distributed particulate contaminant that by now has been detected in various environmental samples (incl. deep sea and the polar regions of the Earth), food and recently also humans. One of the major research activities in the topic of microplastics is towards and standardization and harmonization of existing analytical approaches. That being said, this work clearly falls within an up-to-date topic. However there are a few fundamental concerns that in my opinion prevent this manuscript to be published in its current state.
i) The overall manuscript reads more like a report rather than a high quality and impact research manuscript. It contains outdated literature, for example, the authors claim that only 1% of the plastics have been accounted for and the rest is missing. That information is clearly out dated or at least does not cover the whole story (for example, https://iopscience.iop.org/article/10.1088/1748-9326/aa9500).
ii) Another point this direction is the number of papers that are cited. Of course this should not be the sole criteria but point to the superficial character of the work. This is especially relevant since the analytics of microplastics is one of the few aspects with a considerable development over recent years.

Experimental design

iii) The authors use NaI for the density separation of the microplastics from the sediment and obtain nearly 100% recovery for almost all of their samples. It would be interesting to consider the densities of the polymer particles and take this into account for the discussion of the obtained results.

Validity of the findings

iv) I am missing more objective discussion of the limitations of the developed device. In comparison to glass funnel, where only very small amount of sediments can be used, the here developed device can handle a variety of different sample amounts (l. 180-185). While this may be true, looking at the device in its current form only comparably little sample amount will be handled by this device. Given the so-far expected small concentrations of microplastics in sediments, the sample volumes of orders of magnitudes higher will have to be extracted to reach robust results in the fields. This clearly limits the applicability of this technique.

Additional comments

Finally, although not being a native English speaker myself, I would highly recommend the authors to have their manuscript proof-red. Some of the sentences are somewhat difficult to understand or seem grammatically incorrect (examples are in lines 22, 26, 34 etc).
I hope the authors find my comments useful to further improve the quality of their manuscript.

Reviewer 2 ·

Basic reporting

Minor comments:

1. The authors should improve the English language throughout the manuscript to ensure that an international audience can clearly understand the text. Some suggestions can be found below:
L. 21: "on" the ocean's surface
L. 25: write "resuspension" instead of "re-suspension" throughout the manuscript
L. 34: "This new device is easy to clean and portable, yet enables rapid separation of microplastics from sediments, and these microplastics remain uncontaminated so can be used for further chemical analysis." (divide this sentence into two sentences for better understanding)
L. 38: well investigated
L. 42: , which are referred to as "the missing plastics".
L. 44: locations
L. 45: about how much plastic is accumulating in ocean sediments,
L. 58: onto
L. 65: "through" is not a suitable word in this context
L. 117: use "sand" instead of "soil"
L. 119: "during" instead of "in"
L. 120: that there were
L. 172: "low" instead of "lower"

2. Regarding the title: why is the method only suitable for marine and not for freshwater sediments?

Major comments:

3. The introduction is definitely too short and lacking important information.
Please, define microplastics (MPs).
L. 43: Are there just fragments and fibers in the deep sea? What about films, spheres??
L. 47: How could your method help in investigating the impact of MPs on marine animals? Also, please provide more references from the literature here on MP effects on animals (e.g. the ones living directly in or on the sediment).
L. 50: When introducing current methods for MP separation from sediments, you are missing important references: e.g. Felsing et al. 2018. A new approach in separating microplastics from environmental samples based on their electrostatic behavior. Environmental Pollution.; Grbic et al. 2019. Magnetic Extraction of Microplastics from Environmental Samples. Environmental Science & Technology Letters. There are also some recent reviews dealing with this subject which you could cite.
Fig.1: legend: f: microplastics in the supernatant in the upper tube are poured by rinsing the internal walls of the tube.

Fig.1: I suggest to exchange the positions of pictures a and b so that the first picture shows the upper and lower plate separately.

Fig. 2: write "polyethylene" instead of "polyethene"; "µm" does not have to be written in italics.

Fig. 3: I do not understand what you mean by your y-axis labelling. Also, percentage is misspelled. I suggest labelling the y-axis using the term "recovery rate". Please, include the detailed grain sizes in the x-axis labelling.

Experimental design

LL. 54-57: Name the densities of the different salt solutions and why they are suitable for extracting MPs. Also, refer to the MP densities.
L. 64: " There is an upside-down funnel-shaped microplastic separator with a shut off valve (MPSS), but it is a large metal device and is thus heavy and expensive (Imhof et al., 2012)." Please, rephrase that sentence. Being a large metal device is not a justification for being expensive.
L. 67: It is SMI and not MSI.
L. 93: why did you choose a NaI solution? Again, refer to the MPs' densities here.
L. 96: You write that you used a magnetic stirring bar to mix the sediment for 20 min. I would like to know which type of magnetic stirring bar you used for that. From my experience, most magnetic stirrers possess a plastic (PTFE) coating which can easily be abraded by the sediment grains during mixing. Hence, you should find a lot of small plastic fragments from the magnetic stirrer contaminating your sample when mixing the sediment sample with the stirring bar.
L. 101: Did you also filter your distilled water before using it to remove any contaminants like you did it for the NaI solution?
L. 102: Did you cover the Petri dish with aluminum foil as well?
L. 114: why did you choose those MP sizes? Are they the most common MPs in deep sea sediments?
L. 117: did you really grind the sand?
L. 119: "These added microplastics were distinctive, both in color and shape, ensuring that only the added plastics were counted in the final enumeration." Which colors and shapes did they possess so that you could easily identify them as plastics by eye? You could provide photos. Also, MP verification by µFTIR analysis would have been useful.
L. 120: Which preliminary experiment are you referring to? Please, describe it in more detail.
L. 125: Again, which magnetic stirrer was used? Did mixing the sediment with the magnetic stirrer reduce the size of the added microplastics through microplastic abrasion?
L. 128: where did you collect the reef carbonate sediment and how did you grind it into fine, medium and coarse fractions?
L. 130: provide information on PVC density.
L. 132: are you again referring to wet weight when speaking about the sediment mass? If so, then please write that into brackets.

Validity of the findings

LL. 135-139: How did you test for normality of your data before performing the ANOVA?
L. 143-147: Move that section so that it appears after your results.
L. 151: You are referring to polymer density here but throughout your whole manuscript, you are not introducing the plastics' densities once. Please, provide information on the densities of the plastics that you used for your experiments.
LL. 151-152: Please, explain your statistics again in more detail (e.g. name the different levels analysed by the One-Way ANOVA)
LL. 151-152: write "particles" instead of "particle"
LL. 164-167: Please, discuss why there was no difference between the different sediment types (fine, medium, coarse).
L. 167: "." after "(Fig. 3)"
L. 216: Please, explain which contamination is avoided here.

Regarding the Supplementary data file:

1) PE is polyethylene and not polyethyrene.
2) You are providing MP collection rates in supplementary data 1 as ratios and in supplementary data 2 as percentages. I suggest using percentages throughout the whole data file as it should be consistent. When looking at the supplementary data for PE (B9) you write that you collected 38 MPs out of 40 MPs which would mean that you collected 95 % of the MPs with which you spiked your sample. However, when presenting the data you write that that equals 1.0 (100 %) which is quite misleading as, of course, only 40 collected MPs would mean that you recovered 100 %. Please, change that to avoid any misunderstandings. Also, you report the same collection rate of 0.9 for a recovery of 37, 36 and 34 MPs out of 40 MPs. Of course, the recovery rate is lower if you collect 34 or 36 MPs compared to your recovery rate if you collect 37 MPs. Please, adjust the precision of your calculations throughout the whole data file.
3) Do you mean PVC instead of polystyrene for supplementary data 2??
4) There are often graphic symbols together with Japanese symbols in the data file. Please, change that so that the readers can fully understand your data.

Additional comments

The authors present a small device which they developed to facilitate the separation of microplastics from marine sediments. Although I can see that the device they propose offers some advantages, such as easier cleaning and the prevention of clogging of the device by larger sediment grains, I do have some major concerns regarding the manuscript. Most importantly, the manuscript is very short and only insufficiently addresses the current developments in microplastic research. This is corroborated by the limited number (only 10 references!!!) of references from the literature. Furthermore, the authors should look at their data analysis again. In my opinion, the method they are presenting should rather be published as a Technical Report or Short Communication.

---

## Round 0.2 · Minor Revisions

We thank you for considering the reviewers' suggestions seriously. Please consider a few considerations further regarding e.g. statistical analysis.

Reviewer 2 ·

Basic reporting

The revised manuscript clearly shows that the authors have taken the reviewers' comments into account. They have improved their manuscript by adding recent literature and by proof-reading their manuscript.

Experimental design

The experimental design has been well described by the authors.
However, there are still three things (e.g. regarding the statistical analyses) that should be addressed by the authors before the manuscript can be published.

1) First of all, what do the authors mean when saying that the 30 g is the "wet" weight of sand in L. 151 in the revised manuscript? Was the sand soaked in NaI solution?

2) Do the authors really mean "mean extraction efficiencies" in L. 177? I assume that they used all the data on MP recovery rates (from replicate 1 to 5 per polymer type) for their analyses/ANOVAs.

3) Also, the authors should report the results of their ANOVA before presenting the results of their post-hoc tests in L. 181 and 182.

Validity of the findings

The authors have successfully clarified their data analysis in the supplementary material, but they still need to clarify their statistical analysis (see comments).

Additional comments

The revised manuscript clearly shows that the authors have taken the reviewers' comments into account. By adding recent literature and clarifying their data analysis the manuscript has been significantly improved. However, there are still three things that should be addressed by the authors before the manuscript can be published (see comments). Once these issues have been addressed, I am convinced that the manuscript can be published.

---

## Round 0.3 · accepted · Accept

Thank you for addressing the concerns of the reviewer.